# Performance, Art, Institutions and Interdisciplinarity

**Rob Gawthrop**

Independent Researcher, Sheffield S11 9BY, UK; rob@robgawthrop.co.uk

**Abstract:** How have funding, art education, and politics affected the development of performance and interdisciplinary art? In England in particular, performance as an experimental and radical art practice developed largely from underground activities, political action and a range of art forms. Funding bodies, colleges and art institutions eventually accommodated, albeit to a limited extent, this activity. As financial circumstances were sometimes difficult, artists often provided their own support structures and organisations. Some of these became established as they became successful. Performance art split from the theatrical and became defined as live art. In more recent times, conditions shifted again, and critical, experimental, or avant-garde theatre, film, music, etc., found refuge within contemporary art. Performance however, became increasingly confined and restricted by: the regulatory and academic requirements within universities; the need for evidence for some form of public or social purpose by funding bodies; and the increasingly hostile social and political circumstances. This research draws partly from personal experience and reflects on cultural conditions since the 1970s.

**Keywords:** live art; UK; late 20th century; avant-garde; post-disciplinary; contemporary art; education; politics

## 1. Introduction

Following the Second World War, in response to rationing, austerity and prevailing conservative beliefs, cultural attitudes changed. Continental philosophy challenged modernity's ideas of progress, immutable truth and Eurocentrism. In 1947, Max Horkheimer and Theodor Adorno published "The Culture Industry: Enlightenment as Mass Deception" in *Dialectics of Enlightenment* (Horkheimer and Adorno 2002, p. 94) and twenty years later, Guy Debord in *Society of the Spectacle* wrote: "Everything that was directly lived has moved away into a representation" (Debord 1970, para. 1). In the 1950s, American rock and roll, beat poetry, free jazz, underground film and abstract expressionism were particularly influential among young people in the UK, supplanting folk, skiffle, trad jazz and kitchen-sink painting. In *Bomb Culture*, Jeff Nuttall, himself a poet, artist and performer, considered this period of change and wrote: "between the autumn of '67 when I completed this manuscript, and the summer of '68 when I am writing this preface, young people, under various pretexts, made war on their elders, and their elders made war on them. The war continues" (Nuttall 1970, p. 7).

Opposition to the atom bomb, the Vietnam War, Soviet occupations, censorship, and inequalities produced protest movements and demonstrations among students, trade unionists, and the intelligentsia. It is from this cultural background that live, time-based and experimental art emerged.

Focusing on England, this article addresses how arts education, funding and politics affected the development of performance and interdisciplinary art in the late twentieth and early twenty-first centuries. To avoid a deterministic narrative, a nonlinear structure of broadly thematic and inter-related sections has been used to reflect on the circumstances of the period. A brief outline of historical and social contexts is given in *Background*, and the changing nomenclature of the fields and genres of art and their effect on practice is explored in *...Art. State Support* outlines how political ideologies have impacted

culture through the changing forms and amounts of public funding available, while ***DIY Independence and Collaboration*** outlines the development of arts centres and artist-run organisations and demonstrates approaches that performance and interdisciplinary artists have taken in response to the varied and changing funding regimes. ***Magazines*** charts the development and decline of publications that promoted experimental and radical practices, while ***Television*** outlines the original remits of BBC2 and Channel 4 and their arts programming since. ***Structures and Development*** concerns the reforming policies and structures of national, regional, and non-governmental arts organisations that fund or support live and time-based art. How faculties, courses, and pedagogy in tertiary fine art education have adapted in response to successive higher education Acts of Parliament and budget statements is covered in ***Art School***. ***Practice and Research*** follows on with how universities and other supporting institutions have influenced artists and their practices. Bringing the research closer to the present, ***21st Century Conditions*** briefly outlines the technological, political and social conditions impacting the arts this century. ***Summary*** reflects on the changing circumstances for interdisciplinary and live art since the 1970s and ends with ***Not Concluded***, which offers some analysis and speculation for the present and future.

## 2. Background

The history of performance (or live art) has been largely centred on its significations, expressions and appearances. Academic and theoretical texts followed. In the UK, there has been limited research into how and why performance work from the late 1960s developed, proliferated, or, since the late 1990s, declined. Publications that focused on performance inadvertently charted the changing nomenclature and grappled with locating work that could now be described as post-disciplinary. Initially, support from galleries, funding bodies and art schools was at best, with some exceptions, passive but more frequently hostile.

Preceding the contents page of Adrian Henri's 1974 *Environments and Happenings*, Allan Kaprow's 'Definitions' states:

> The term 'environment' refers to an art form that fills an entire room (or outdoor space) surrounding the visitor and consisting of any material whatsoever, including lights, sounds and colour.

> The term 'happening' refers to an art form related to theatre, in that it is performed in a given time and space. Its structure and content are a logical extension of environments.

Also in Henri's book, the first chapter, *Towards a Total Art*, lists eight disparate events and writes, "The common quality all these activities share is their awkwardness, their inability to fit into a preconceived artistic framework". (Henri 1974, p. 7). Similarly, Rosalind Krauss in 1977 stated that "We are asked to contemplate a great plethora of possibilities in the list that must now be used to draw a line around the art of the present: video; performance; body art; conceptual art; . . . characterized, now, not by rigor but by a wilful eclecticism.. . . (ellipsis in original) Both the critics and practitioners have closed ranks around this 'pluralism' of the 1970s" (as cited in Sayre 1992, p. xi). Within modernism, art was a progressive, critical, and autonomous practice. The physical manifestations of modern art—the art object, exhibited in galleries, progressive but becoming commodified—were to eventually serve an international art market. The conflict between integrity and making a living produced compromises and contradictions. Since the 1950s, the move of some artists away from object-making towards performing or exhibiting in the public realm has been motivated by an antipathy towards the commodification of art. In 1968 Douglas Huebler stated that "The world is full of objects, more or less interesting: I do not wish to add any more" (as cited in Celant 1969, p. 43). The desires to engage with a broader public, to discover new forms and to question the status quo were all important factors. Examples include *Happenings* in the USA, *Material Action* in Austria, *Actionism* in Germany, dissident underground actions in Poland, Hungary, former Czechoslovakia and Yugoslavia, and

*decollage* and *situations* by Lettrists and Situationists in France. *Fluxus* artist Dick Higgins in 1966 proposed "The Intermedia" (Higgins 1993, pp. 172–73); Henry Flynt coined the term "Concept Art" in 1963 (as cited in Lippard 1997, p. 258); and in 1960, George Brecht described "the total, multi-sensory experience that could emerge from a *situation*, the *event* being the smallest unit of a *situation*" (as cited in Stiles 1993, p. 66). These were concerned with *experiences* where time and process were fundamental. Through the use of *scores*, anyone was permitted to enact particular works.

The UK seemed to have been rather apart from such activities, with performances largely deriving from street and fringe theatre, including Welfare State, The People Show, etc., or associated with the rock music scene of the swinging sixties, when artists including Mark Boyle, John Latham, Joan Hills and others worked with a number of bands, including Pink Floyd and Soft Machine. Gustave Metzger (who influenced the Who's performances by smashing up their instruments on stage), John Latham, Jeff Keen, J. G. Ballard, Fluxus artists Yoko Ono, Wolf Vostel and Al Hansen, plus Viennese Actionists, were all part of the *Destruction in Art Symposium* in London in 1966. Around that time, Barbara Steveni founded the *Artists Placement Group* (APG) along with John Latham, Stuart Brisley, Ian Breakwell, Anna Ridley and David Hall. APG was an artist-run organisation that worked outside of galleries and placed artists within institutions and industries. The Fluxus ethos of intermedia had entered the UK's alternative cultures and the avant-garde. At St. Martin's School of Art (1963–1966), there was a rejection of the type of abstract metal sculpture—promoted by Anthony Caro and Philip King—by students, including Richard Long, Bruce McLean and Gilbert & George, who at the time were producing conceptual, transient and performance works. Things changed with the appointment of Peter Kardia, "who in 1965 devised a single Fine Art programme, combining the painting and sculpture departments for the first time. Though this was initiated as a strategic convenience to satisfy national diploma and degree requirements, the programme immediately resulted in a radical shift" (Le Grice n.d. [online] para. 7). The established art scene was still mainly concerned with the forms of modernism exemplified by Henry Moore, Barbara Hepworth, Lucien Freud, and artists associated with St. Ives, Cornwall. The mainstream media was still ridiculing Picasso, Pop Art and the abstract expressionists even into the 1970s. In 1976, Carl André's *Equivalent VIII* at "The Tate was ridiculed by many for, as they saw it, being conned into buying a 'pile of bricks'" (Tate n.d. [online] second subsection). Public art events were often regarded as publicity stunts or acts of outrage (I witnessed members of Nice Style being physically threatened by a journalist in 1975 at their Deep Freeze performance). Conceptual art, Art Povera and other emerging movements only began to come to the fore following the exhibitions *When Attitudes Become Form* came to the ICA in 1969, *The New Art* at the Hayward Gallery in 1972 and the opening show at Gallery House in the same year, which included Marc Chaimowicz, Bruce McLean and Stuart Brisley.

The art magazine *Studio International*, under the editorship of Richard Cork in 1975, became an advocate of the new art and published Special Issues including *Avant-garde Film in England & Europe* (1975), *Art & Experimental Music and Performance Art* (1976). There were also review sections: *Film* by Malcolm LeGrice, *Video Art* by David Hall, *Sound* by Michael Nyman, *Performance* by Marc Chaimowicz and others. This was the only publication in the UK at the time that was concerned with the emerging discourses around art, culture and post-modernity. *Studio International* was eagerly read by art students and their more progressive tutors. Mainstream critics and many art school tutors were antagonistic to such ephemeral and temporal works and rejected them as not being art at all. Reflecting this reaction against the *new art*, the journal *Artscribe* was founded in 1976 "by a group of painters and critics . . . who were discontented with existing journals that concentrated on conceptualism, socialist art, new media and academic theory" (Walker 2002, p. 178). This had the objective of reaffirming painting at the top of their artistic hierarchy. Performance, sound, film and video as avant-garde practices were still peripheral to art. The tensions between the disciplines that they emerged from and the art world that was being challenged

gave these practices a particular potency, as they were not dependent upon galleries and were presented in cinemas, on stages, on public sites and in empty spaces.

## 3. . . . Art

Whereas *intermedia* had to do with work that accommodated, crossed, or combined different art forms such as music, art, or dance, interdisciplinarity tended to be concerned only with crossing *art* disciplines. Performance, almost by definition, was intermedia, as music, dance, poetry, etc., can all be performed, and these art forms had their own avant-garde. Such avant-garde work would not necessarily be distinguishable by its art-form provenance. For funding bodies and colleges, these areas of practice did not conform to their defined departmental structures. Physical theatre, street theatre, post-modern dance, body art, expanded cinema, sound poetry and performance art were all terms that sought to define areas of practice based on the discipline that the work had stemmed from. These areas of arts practice, however, had the shared characteristic of temporality, and in the 1980s, the term *time-based* began to be used and usually suffixed by either *art* or *media*. The division between *art* and *media* could include or exclude performance, and in the case of time-based media, had an explicit link with technology. The common ground for performance art was activity in proximity to an audience, being of the body and the sensible (as opposed to the intelligible) and a resistance to conform to existing structures and methods. The defining field of concern therefore would be *performative*, which, by way of the use of materials, its documentation, or means of transmission, is inescapably interdisciplinary.

Performance in the first instance is *live*—that is, being experienced by an audience at the same time and space as the actions of the performer or performers. It can also be mediated through broadcast, streaming, live-feed, or, recorded and documented through text, drawing, photography and film. Such documentation may supplant the original event and achieve the status of its own as an art object.

As performance art in general included conventional work such as musicals, circuses, recitals, etc., some form of definition became necessary. In 1986, the Arts Council of Great Britain (ACGB), as it was, formed the *Combined Arts* panel, and within that, the *Performance Art Advisory Group*, which, in turn, became the *Performance Art Promoters Scheme*. The term *live art* was already being used and an Arts Council publication of the time stated: "Performance Art is a term used to describe a live art activity that fits uneasily into critical categories and has an ambiguous, if not controversial place in the public mind" (Walwin 1987, unpaginated). *Live* art in relation to being recorded is a relatively recent term, as prior to this, *live* was just used as the opposite of dead. "*Live* as we apply it to music (and potentially to all face-to-face communications) entered the lexicon of music appreciation only in the 1950s . . ." (Sterne 2003, p. 221). This was largely because of the wider availability of audio and video recordings. In respect of performance, a distinction could be made between the documented and *live* as as *description*. As a performer, it makes little sense to call oneself a live artist. Nevertheless, in 1991, the Arts Council replaced *Performance Art* with *Live Art*, removing the ambiguity that performance had in relation to dance, music, etc., but affirming potential interdisciplinarity within art (rather than arts). It was therefore incumbent on organisations and artists seeking funding to adhere to this terminology. Another ambiguity relates to work that is live broadcast (radio or television) or live streamed (web-based or live feed). It can be said that any live performance in this respect would be *mediated* and that occupying different spaces would separate audience and artist. The simultaneity of action and its reception would be largely dependent on trust. This is comparable to the *acousmatic* in sound theory, where visual identification of the sound source is denied. Jonathan Sterne traced the origins of mediation to medicine and the invention of the stethoscope. When sounds of the body were listened to directly (auscultation), it was *immediate* (nothing in between), but listened to through an apparatus, sound becomes mediated. "Mediate auscultation is the physical, spatial configuration of a particular knowing" (Sterne 2003, p. 107). The media that communicates a live performance

becomes integral to the event and a particular *knowing* becomes necessary, or such an event would be experienced as documentation, effectively denying its *liveness.*

*3.1. State Support*

State sSupport for the arts began to slowly develop following the election of a Labour government in 1948. Alongside the creation of the welfare state and, the nationalisation of power and transport, the government shifted the ideological centre to create a post-war economic and social consensus of social democracy. The Arts Council of Great Britain (ACGB) was created and a later Labour government in 1964 appointed the first Arts Minister. The first arts policy document (the White Paper) was published soon after and the Regional Arts Associations (RAAs) were set -up. Funding began to be available more widely for artists and arts organisations. Traditionally, artists had earned a living through sales of their work. Public galleries were state-subsidised to enable them to purchase works, whereas commercial galleries received a percentage from sales. Conceptual, time-based and live art challenged these funding models. Changes happening internationally regarding art were slowly gaining recognition by funding bodies, galleries and art schools and artists began to receive payment for performing or presenting work that was, by its nature, unsalable. The ACGB funded national initiatives and the RAAs funded artists and arts organisations' programmes. Local authorities supported their own museums, galleries and other arts institutions, and small funds were sometimes available to artists and local organisations.

The political consensus ended with the election of Margaret Thatcher's Conservative government in 1979. Swingeing cuts were made across the public sector. Although arts funding was reduced, the arms-length approach continued. Performance art was sometimes controversial and the media often gained much mileage from this, frequently stirring-up outrage and fuelling the prejudices of right-wing ideologues against public funding of such work. The Daily Mail's front-page headline quoted Conservative MP Nicholas Fairbairn's 1976 comment on COUM Transmissions' *Prostitution* show at the ICA that "Public money is being wasted here to destroy the morality of our society. These people are wreckers of civilisation" (as cited in Reynolds 2006, p. 229). The attitude that experimental and avant-garde art should not be publicly funded was (and still is) typical among Conservative politicians. The attitudes of local government officers sometimes succumbed to such media pressure. In fact, currently all arts and humanities are being attacked, particularly within universities, with courses threatened with closure. The ACGB, being an arms-length (but government funded) organisation, and most RAAs were generally supportive and resilient against such opposition. It was in 1987 that the government demanded a restructuring of the ACGB's funding. A review was undertaken and in 1990, the RAAs were reconfigured into Regional Arts Boards (RABs) with quasi-autonomous status. Organisations and individuals applied to the RABs and not the ACGB. In 1994, the ACGB was split into national arts councils; England (ACE), Wales, and Scotland (Northern Ireland already had its own system) and it was planned to eventually abolish the RABs. The National Lottery was also established in 1994 and funds were raised for *good causes*, including the arts. Arts lottery money was distributed by the government via the national arts councils and the British Film Institute. This militated against further cuts by the government. However, there iswas now a demand for artists and arts organisations to seek private funding and corporate sponsorship when making an ACE or RAB application. The government, during its eighteen years in power, also introduced various schemes to encourage entrepreneurship and take people off the unemployment register. Many artists used unemployment benefits to survive but also capitalised on schemes such as the *Enterprise Allowance* to support their practice as a business buying materials, equipment and gaining access to premises. The destruction of manufacturing, particularly in docklands and the manufacturing centres of the north, left many industrial buildings and warehouses empty, and these were obtained by groups of artists and organisations either legitimately renting or squatting.

In 1997, Labour returned to power and formed the Department for Culture, Media, and Sport (DCMS), which for the first time included a senior cabinet post. In 1998, Regional



Development Agencies (RDAs) were created by the government to deal with and distribute the European Regional Development Fund (ERDF). A proportion of this fund was used for regeneration and it was recognised that the arts had a role in this. Gateshead Council's commissioning of Anthony Gormley's sculpture *Angel of the North* in 1998, on a hill above the A1 road, was a powerful symbol of such social intervention. Funding was also made available through ACE for large capital projects, enabling organisations to develop new centres for studios, facilities, events and exhibitions. This increased funding was also coupled with a social agenda regarding equality, diversity, and health. In 2001, the RABs were merged with ACE, which enabled artists and organisations to work directly with regional ACE officers. ACE also devolved some funds to arts organisations and galleries as *Regularly Funded Organisations* (RFOs) (currently named as the ACE national portfolio) that enabled them to develop projects, commission artists and fund artists themselves. The demand to obtain matched funding, however, continued. New UK border controls were introduced following the formation of the UK Border Agency in 2008. Non-EU artists, performers, lecturers, etc., needed to have sponsors to legitimately enter the country. Visa requirements for artists from many overseas countries made travel to the UK difficult and, in many cases, not viable.

In 2009, a Conservative and Liberal Democrat coalition government was elected and imposed an austerity budget. The Border Agency became the Border Force and the *hostile environment* policy was introduced, imposing further restrictions on entry to the UK. The government grant to ACE was cut by 30% in real terms over ten years. ACE centralised its activities, reducing infrastructure costs in an attempt to safeguard its funding commitments. This resulted in reduced access to ACE officers and an increasingly systematised application process. Government contributions to local authorities was reduced on a year-on-year basis, with some authorities suffering cuts of more than 50%. The COVID-19 pandemic, Brexit, changes of (Conservative) prime ministers, twelve culture ministers in thirteen years, the Ukraine war, the cost-of-living crisis and industrial unrest have left the arts in a state of crisis, with organisations closing and prospects for artists becoming increasingly uncertain. The end of EU *Freedom of Movement* imposed new restrictions on travel by artists to and from mainland Europe.

### 3.2. DIY Independence and Collaboration

The cultural climate in the late 1940s was clouded by the Second World War, and state galleries were still concerned mainly with the work of war artists and old masters. John Rothenstein, the former director of the Tate gallery, wrote in 1974 that the *Contemporary Art Society* (founded in 1910) purchased a Francis Bacon painting in 1946, this "was [made] available as a gift, to almost any public collection, yet six years passed before it found a home—in the Bagshawe Art Gallery in Batley, Yorkshire, one of the obscurest in Britain" (as cited in Gregory 2021 [online] 9 February 2021 para. 8). Modern art, and abstraction in particular, was perceived as something shocking. Many avant-garde artists of the time found Yorkshire and Cornwall more welcoming than London. It was possibly this fact that led to the formation of The Institute of Contemporary Art (ICA) in 1946 by a group of artists, critics and patrons (including members of CAS), which, in 1948, opened its first exhibition. The ICA was founded "to position itself at the forefront of art and culture. Initially conceived with a special focus on artists working across a range of contemporary art forms . . ." (ICA n.d., History [online] para. 1). Other arts centres developed; formed by collectives of arts practitioners, funded by the ACGB and supported by local authorities. Significant for performance in the 1970s, along with the ICA and others, were The Midland Group (Nottingham), Arnulfini (Bristol) and Chapter (Cardiff). Such arts centres in the late sixties and seventies accommodated what had been called the *counter culture* and became venues for experimental and avant-garde film, music, and performances. In London, the Covent Garden market closed in 1974 and a number of empty buildings became temporary art galleries, including ACME, AIR, The Garage and PMJ Self. In London's Docklands, empty warehouses were taken over by a range of cultural practitioners and 2B Butlers Warf

was significant for performance and video between 1975 and 1978. These spaces enabled a cross-section of artists, musicians, and performers to develop work in situ, perform one-offs and construct installations. Many municipal galleries also allowed such events to take place where established theatres, concert halls and cinemas were inaccessible (*Coum Transmissions* performed at the Ferens Gallery, Hull, in the early seventies). This period also saw the formation of artists' collectives to share studio space or to provide film and video production facilities. Many of these artists' groups were formed by recent art graduates and it was arts graduates and students that made-up a large proportion of audiences.

As political, social and cultural circumstances changed during the mid-late seventies, the confluence of an excluded working-class youth with a new generation of art students who had a *punk* DIY ethos emerged. Exhibitions, events and performances developed outside of established systems in warehouses, artists' studios, film workshops, sound/music spaces, etc. Throughout the 1980s and 1990s, art students, with support from staff at a number of art schools (including Coventry, Hull, Maidstone, Cardiff and Sheffield), developed what was generically called *events weeks.* These operated on an open selection basis for students to perform or show their work. Artists were also invited and paid through visiting lecturer fees, the Arts Council *Film & Video Artists on Tour and the Performance Art Promoters Schemes*. In 1979, the first *National Review of Live Art* (NRLA) was held at the *Midland Group Arts Centre* in Nottingham and the *Basement Group* was set -up by art school graduates in Newcastle (later becoming *Projects UK* and then *Locus+*). *Transmission Galley* in Glasgow (set up by art school graduates dissatisfied with the lack of opportunities in Scotland) and *Hull Time-Based Arts* (HTBA) were founded in 1984. Projects UK and HTBA were the first artists' groups in the UK that concentrated on presenting interdisciplinary work in publicly sited spaces.

In 1988, *City Racing* was established in a derelict betting shop in Brixton, South London, by a disparate group of artists who had previously graduated from art schools across the country. The model was similar to Transmission's, i.e., giving space to any artist proposing work that appealed to the organisers. "The truth was that every show was determined as much by accident and coincidence as by curatorial steel" (Hale et al. 2002, p. 5). Though it was never stated, none of these organisations made distinctions regarding art forms or disciplines. The first *Edge Biennale of Experimental Art* took place at various sites in London in 1988, Newcastle in 1990 and Glasgow in 1992. This followed a series of performance events at the AIR gallery organised by Rob Le Frenais (who, after curating *Edge*, was curator for *Arts Catalyst*). *Edge 88* was a collaborative event including AIR Gallery, Projects UK and Performance Magazine, of which issue 55 was also the catalogue. It was also in 1988 that students and graduates (mainly from Goldsmiths College of Art) staged *Freeze* in a derelict building in London's Docklands (these artists were later known as the *Young British Artists*, or YBAs). *Frieze* magazine was launched as an international glossy that effectively promoted the YBAs. "In February/March 1992 we [City Racing] put on a show of new works by Sarah Lucas [who was included in *Freeze*]. Charles Saatchi turned up and bought three of Sarah Lucas' large photocopied *Daily Sport* pieces for £3000 pounds. This was the first time we had to deal with a collector. We did not have a price list" (Hale et al. 2002, p. 33). This was significant, as previously there had been few collectors of contemporary art in Britain. Commercial galleries recognised the emerging market for British contemporary art and sought out work, particularly by YBAs. The separation between contemporary art curated for an art market and that funded through state and local authority grants for public access became more acute. Documentation of performances and some video art became reconfigured for conventional gallery exhibitions (digitised, looped and listened to on headphones). Avant-garde and documentary film, experimental theatre, music, dance and even architecture found a haven within *Contemporary Art* as the commercial imperatives of theatres, cinemas and concert halls forced unprofitable work out. Public funding for such work did become available, particularly as the Arts Council took on a social agenda. Live, site-specific, socially engaged and time-based art continued to be reliant upon public, charitable and other sources of funding.

Publicly funded organisations tended to provide open submissions or commissions of works, projects and residencies. *Projects UK/Locus+*, *Edge*, *NRLA*, *Video Positive*, *Nosepaint/Beaconsfield*, *Artsadmin* and *Artangel* operated in this way. HTBA's annual *Root Festival of Live and Media Art* began in 1992 with a partnership with The Ferens Art Gallery's newly constructed live art space and with Humberside Dance Agency. HTBA itself was formed by artists, filmmakers, performers and musicians, its director, Mike Stubbs (1997) wrote "Some of the members went to art college. Others are people who, feeling disenfranchised as either artists or audiences by the mainstream, get involved because HTBA makes the marginal permissible" (Stubbs 1997, p. 00.30). "Their explicit social and political imperative positively engaged with those who had been under represented in the past. The constitution is notable in its aim to promote local work alongside international artists" (Wilczek 2017, p. 124). With some exceptions, most of these organisations, festivals and events were outside of London, where there were few existing audiences, art collectors, *national* museums, or galleries. This meant that organisations could not take anything for granted and had to undertake all of the work themselves. These organisations—cooperatives, charities, collectives, etc.—were not businesses but sought to make marginal work publicly available and give opportunities to known, unknown, local, or international artists. In short, to develop a culture.

### 3.3. *Magazines*

"With its maverick and punk ethos Performance Magazine embodied an immensely active community of artists, writers and publics that crossed disciplines throughout the late 70s, 80s and the start of the 90s" (Performance Magazine Online n.d. [online], home page). It was effectively replaced by *Hybrid*, *The International Cross-Artform Bi-Monthly* 1992–94 and from 1994 to 2003, *Live Art magazine/Listings*. There was also the Glasgow-based Variant (1984–1994). These magazines were important as interdisciplinary journals in disseminating information, reviews, theoretical discussion and documentation. Additionally, specific issues served as programmes and catalogues for festivals and events. Magazines, including *Performance*, *Undercut* (experimental film and video) and *Musics* were mainly media-specific. Small independent journals such as *P.S.* and *Readings* were more interdisciplinary but short-lived. *Art Monthly*, founded in 1976, has successfully continued to publish without a break. This has been because it has offered a critical view of all aspects of contemporary art, with an emphasis on the UK. In the mid-1990s, journals including *Coil*, *Mediamatic* (Amsterdam) and *Mute* were founded to report, document, and discuss the emergence of art in relation to digital and *new media*. These were the last printed art magazines that had a specialised and public-accessible purpose. New specialised academic journals, both printed and online, were developed in response to the new university research agenda, with *Performance Research* (Routledge) and *Moving Image Review & Art Journal* (Intellect Journals) being examples. These were centred on an analytical or theoretical basis, peer-reviewed and available mainly through university libraries or by subscription.

### 3.4. *Television*

"In 1964 BBC2 was launched with the remit of offering an alternative and more experimental style of television broadcasting" (BBC [online], para. 4). In a letter to Charlie Chaplin, the new BBC2 controller, Michael Peacock stated, "We will be taking risks, experimenting, escaping from the tyranny of ratings and fixed schedule patterns; above all, trying to give programmes the length on the air which they need to develop their full potential instead of forcing material into pre-determined twenty-five or fifty-minute 'spots'". (as cited in Medhurst 2017 [online] p. 15). This approach reached a peak by the mid-1970's but after that, the risk-taking and experimentation declined. Channel 4 was founded in 1984 "to champion unheard voices . . . to innovate and take bold risks" (Channel 4 n.d. [online] 1st & 2nd sub-headings). Similarly to BBC2, this approach gradually became peripheral to its more popular programming. During these periods, there were regular programmes of experimental music, poetry, film, dance, theatre, world cinema, performance and contempo-

rary art. Programmes, however, were usually compromised by their late- night scheduling, additional commentary, idents, interruptions or advertisements. Nevertheless, a broad range of work was available to the general public. Despite the large number of broadcast channels now available, experimental, challenging and even non-English language programmes are rare. Although there is now a considerable amount of work available online, it has to be searched for, and quality is often compromised, nevertheless, it is a valuable resource for students and researchers.

### 3.5. Structures and Development

Existing systems of production and distribution were changing and organisations were adapting accordingly. As mentioned before, funding became available for capital projects. Large-scale initiatives such as *Tate Modern* and *The Baltic* (Newcastle/Gateshead) developed and opened as millennial projects. Independent organisations, galleries and studios sought funding for new buildings, refurbishment, or expansion. Some of the more radical organisations became part of the establishment as they reconstituted into charitable companies and became *centres* for their respective purposes and locations. Some failed, as prospective income generation (usually advised by consultants) was found to be unrealistic. The model of organisation generally required by funders is a company limited by guarantee and usually with charitable status.

> The group will consist of suitably qualified directors or trustees to oversee the mission of the organisation and provide governance oversight. The group will be independent of the executive and capable of taking responsibility for ensuring that the organisation's funding agreement with the Arts Council is implemented".
>
> (Arts Council England 2022 [online p. 4])

Boards would usually be made-up of *the great and the good*, essentially a nineteenth century charitable model but with aristocrats and clergy being replaced with business people and directors of other organisations. Ideas of workers' (artists') control, common ownership, co-ops, and collectives were acceptable, provided they remained small and relied on project rather than regular core funding. *The Side Gallery* in Newcastle lost its Arts Council funding in 2023, and the local Labour MP, Grahame Morris, wrote, "We express concern that the decision to cut funding may be based on the Arts Council's prejudice against the Side Gallery's egalitarian collective governance" (as cited in Warburton 2013 [online]). As "The purpose of a co-operative society is to serve the interests of members". (Co-operatives UK n.d. [online]), the Arts Council has on occasions deemed this a conflict of interest, neglecting the fact that members' interests are not for personal gain. An arts organisation that is an incorporated charity tends to have a hierarchical structure with a director (or CEO), administrator, finance officer, marketing manager and curator, plus support staff (depending on scale). The director is answerable to the board, the chief purpose of which is to ensure that the organisation is benefiting the public by carrying out its purpose and that financial due diligence is maintained. Projects and programmes are generally decided upon by the director and curator and not by committee or through open submissions. Since around 2010, few independent, regularly funded interdisciplinary arts organisations continued to operate. Organisations that have continued include the *Live Art Development Agency* (founded in 1999), which had taken on the role of facilitator of live and interdisciplinary work, along with *Artsadmin*, which also has a roster of artists, and *Artangel*, which has become a highly successful commissioning and facilitating organisation for large-scale publicly sited projects. Galleries *Beaconsfield* and *Matts Gallery* have continued as spaces and facilitators for experimental art. All of these are based in London. Outside of London, *Transmission Gallery* still operates the same; *Locus+* continues, albeit in a reduced capacity; and *Arts Catalyst* has moved from London to Sheffield.

### 3.6. Art School

An individual becomes a performance artist by way of a combination of exposure to, and education in such work, often including some practical participation. Initial routes

in the late 1960s/1970s were through fringe and street theatre, performance poetry, art happenings and political actions. Education came out of both theatre and fine art courses. Theatre (including drama) courses were essentially script- and group-oriented (language and narrative) and fine art courses were generally individual, visual and *object*-based (i.e., painting and sculpture). Theatre and art schools were generally separate, having different entry requirements and pedagogies, with two exceptions being Dartington College of Arts and Bretton Hall College of Education (both now closed). In theatre, an awareness of the work of Alfred Jarry, Bertolt Brecht, Antonin Artaud, etc., and in art, Alan Kaprow, Jim Dine, Yoko Ono, etc., prompted a few motivated students to veer away from the mainstream paths of their courses. A few institutions developed reputations for supporting experimental, avant-garde and differing branches of performance.

Art schools (as opposed to art academies and a small number of fine art and art history courses in universities) developed from early 19th-century schools of design to provide craft and technical skills for manufacturing. Many more were founded as schools of art in the 1890s and 1900s. Most students enrolled came from mainly working-class backgrounds with the intention of entering a trade. In 1958, the Conservative government formed the National Advisory Council on Art Education, chaired by William Coldstream (a painter, art lecturer and film editor), and published its report in 1960. The implementation of its recommendations was essentially responsible for the art school as we think of it now. An important part was the recognition that secondary education did not equip the student sufficiently to embark on diploma-level art and design education and that completion of a foundation course was necessary for entry to higher education. Access was available to students based more on their aptitude and creative skills than their academic qualifications. A foundation student's choice of diploma (later degree) course was largely dependent upon their tutor's knowledge and encouragement. Otherwise, a prospective student finding the right course or institution to attend was, to a greater or lesser extent, a matter of chance.

Fine art diploma or degree "courses were based on studio practice (80%) with art history and complementary studies as a minor component (20%). *Studio* was based on the *disciplines* of painting and sculpture and expanded later to include print, photography and sometimes a *third area* to accommodate students working conceptually or performatively" (Italics in original, Gawthrop 2016, p. 38). Such third areas could be problematic, and as Joseph Beuys in 1972 stated, "I don't believe that an art school, which should stress new artistic concepts, should lay emphasis on fixed places to work in the school. That sort of thinking is tied up with the idea of art as a craft, with the work-bench and the drawing-table" (Beuys 1992, p. 888). Ironically, more recently, studio space in many institutions has been lost, but for economic reasons rather than any progressive pedagogy. Unlike in Europe, the USA, and Japan, specific 'intermedia' courses never really developed in the UK, the Edinburgh School of Arts' course being the only one. Most courses operated on the basis of the allocation of studio space, a personal tutor and access to materials and technical facilities. A minor component (general studies), in addition to art history, included lectures and workshops in any number of subjects, including film, philosophy, music and performance. Attendance was not always monitored, as it was presumed students were there because they had a commitment to study. Again, chance played a part in what was offered and was dependent upon the interests of staff. Interaction between staff and students often happened informally (in cafés or pubs), enhancing a creative culture, though students unable to join in (particularly women and mature students with family responsibilities) were effectively excluded. Tutorials and critique could be destructive for students if they faced demands to defend their practice instead of being offered guidance, support and constructive criticism. The academic year was usually divided into three terms of twelve weeks; two weeks of the final term would be for assessments. A diploma or degree course lasted three years. Post-graduate courses were two to three years. The award was given following an assessment of a final year show. This pattern continued until the mid-1990s.

The Conservative governments since 1979 embarked on a continual policy of public sector cuts and privatisations. Staffing, space and the availability of materials were reduced considerably. Peter Kardia, previously at St Martins School of Art, set-up the post-graduate Environmental Media Course at the Royal College of Art in 1972 and offered in its propectus "students requiring extended or mixed media facilities and for those whose work includes proposals for redefinition of conventional fine art boundaries" (as cited in Ding 2014 [online para. 2]). The course was closed by newly appointed right-wing Rector Jocelyn Stevens in 1986. Mergers of art schools and other colleges into polytechnics had been taking place since the early 1970s and in 1988, the polytechnics were removed from democratically elected local authorities and made into quasi-autonomous corporations. After the 1992 *Further and Higher Education Act*, polytechnics became universities and could validate their own degrees. Module- or unit-based programmes were adopted, and two fifteen-week semesters per year were implemented (which included two weeks each semester for assessments). Each module required *learning outcomes* and for every module, these were to be tested against a *quality procedure* adopted from manufacturing. This reduced the amount of time a student had to attend lectures, workshops, etc. or work on their own practice. Post-graduate degrees were reduced to a single calendar year of three (sic) semesters. Student funding had previously been based on a system of means-tested grants and the value of these were successively reduced and replaced with loans. The change to a Labour government in 1992 maintained the status quo but introduced a policy of expansion that aimed for 50% of school leavers to enter higher education. Course fees were introduced in 1998 and paid for with loans and grants. Materials by this time were rarely allocated, and students had to obtain them themselves. This expansion, coupled with the modular testing regime, shifted the emphasis of education away from the desire for knowledge to a perceived need for qualifications. One aspect of the modular system has been *professional practice*. This normally included C.V., website and application writing, budgeting, networking, marketing, placements and professional experience. The possibility of being an artist as a career started to become embedded and expected. Degree shows nevertheless continue to be the final assessment for fine art undergraduate courses. These used to be largely student-run and relatively localised. Marketing departments have become increasingly involved and degree shows tend to adopt the white cube model for exhibitions. "Unshadowed, white, clean, artificial—the space is devoted to the technology of aesthetics. Works of art are mounted, hung, scattered for study. Their ungrubby surfaces are untouched by time and its vicissitudes" (O'Doherty 1999, p. 15). It takes support from staff and some considerable effort for a student to be able to present or perform work outside of this model. Performance, site-specific work and socially engaged practices do not fit easily with the current curriculum and assessment processes.

### 3.7. *Practice and Research*

Artists (as lecturers) have been and continue to be supported to a greater or lesser extent by art schools and university art faculties. However, this support has changed radically since the 1960s/1970s, when most lecturers were part-time and pay was sufficient to enable a significant amount of time to practice. Full-time staff had professional contracts that enabled them to manage their own time with the expectation that an average of one day per week and a proportion of vacation time were for their own practice. The cuts in funding through the 1980s made part-time working less sustainable and a move to convert such hours into fractional but permanent contracts was expedited by teaching unions. Staff instigated, participated in, collaborated on, or supported student projects, including residencies, placements, exchanges, visits, festivals and events. These were often outside of normal working hours and not necessarily within a formal curriculum. The 1992 *Further and Higher Education Act* replaced the professional staff contract with the *National Contract* (UCU n.d.), which prescribed the weekly and annual scheduled teaching, annual leave and time for research and other scholarly activity. This was a compromise that maintained the professional nature but within determined limits.

Formal research and research degrees in arts practice were developed for the first time in post-1992 universities with funding from the *Arts & Humanities Research Council* (AHRC) and the *Research Excellence Framework* (REF). The REF assesses research every five to seven years and distributes funding based on the outcome of their assessment. Arts and humanities research is based on the method of posing questions and researching to seek answers. This, being more teleological than creative, sits somewhat uncomfortably with the practice of art. The research degrees of MRes, MPhil and PhD demand a comprehensively referenced thesis that may include practical work, which is defended by the candidate at a viva voce with examiners. University research works in a similar way in dealing with research questions, usually by way of a theme or through a research group. Funded themes are set by the AHRC and research groups within the institution. Universities are increasingly demanding lecturers to have PhDs and a research profile. Over time, this is likely to affect students' education and the work they may produce. Work that has come out of such cultures can often conform to an instrumentality or be overly academic (in the pejorative sense).

There are considerable philosophical, technical and administrative complexities that staff and institutions have to engage with in order to gain funding. Practical work produced under this research regime can tend to be academic, illustrative, interpretive, or unnecessarily erudite. Exceptions are when those concerned (individually and institutionally) recognise that practice is primary and that the writing is an attempt to articulate and evidence what the practical work has achieved.

Work that presents an audience with a sensory or temporal experience—of the visceral, abject, beautiful, sublime or emotionally challenging; that questions answers or gives a sense of wonder—does not always fit with the research themes that university managers perceive as necessary to gain funding. The REF assesses art research within two Units of Assessment (UoA): "Art and Design: History, Practice and Theory" (UoA 32) and "Music, Drama, Dance, Performing Arts, Film and Screen Studies" (UoA 33); this also includes "performance, live and sonic art". (REF 2021 [online] p. 34). These two *units* reflect differences, not just in terms of media or disciplines, but also between the mute static object and the temporal aural process. Art on one side may be associated with useful or decorative objects and on the other with narrative and entertainment. In Albert Camus' 1952 *The Rebel*, an equivalent split is politicised between bourgeoisie decoration (formalism) and socialist propaganda (realism). "Language destroyed by irrational negation becomes lost in verbal delirium; subject to determinist ideology it is summed up in the word of command. Half-way between the two lies art" (Camus 1971, p. 238).

### 3.8. *Twentieth-Century Conditions*

Technologies that emerged in the 1990s have become part of everyday life, with the smart phone rapidly becoming essential. The internet and social media have facilitated open platforms for music, film and other content, though without much remuneration for artists.

Exiting the European Union has had only negative consequences for artists, students, and researchers regarding funding, travel and the sharing of knowledge and creativity. The COVID-19 pandemic curtailed all forms of in-person collective activity, with many practitioners losing their livelihoods despite the efforts of the Arts Council. Economic austerity, privatisations, erosion of public services and the economic disasters of successive Tory governments caused a cost of living crisis, resulting in much deprivation. The climate emergency escalates without effective action; war has returned to Europe with Russia's invasion of Ukraine; and the outrages in Palestine and Israel have produced a climate of anger and helplessness. The British government continues to bring in ever-repressive legislation against protests, trade unions, asylum seekers and refugees and continues to marginalise arts education at all levels.

## 4. Summary

Performance and live art histories since the Second World War have been caught between modernism (art movements, progress and the avant-garde) and post-modernism (ideas around appropriation, hybridity, post-colonial theory, sexuality, and identity). Performance as part of film events tended to be identified separately as *expanded cinema* and within *experimental* film rather than either art or cinema. Music, though, as *performed*, can be either live or played from recordings. The borders between experimental music and sound art are blurred, with the latter also presented as gallery installations. Modernist notions of progression within established media disciplines tended to be accepted by funders, galleries and other venues and facilitated despite resistance from traditionalists and mainstream media. Though artists were generally at the forefront of post-modern practices, theoretical ideas and concepts tended to be confined within academia and within some magazines and journals. However, magazines and to a lesser extent, television and radio, became the main sources of information, reviews, and debate.

Mainstream media tended to promote the traditional, reflecting the conservative values of maintaining the status quo, the related liberal notions of freedom, or at best, art's *civilising influence*. The Labour Party, recognising the social function of art, supported art that questioned the status quo, though some sectors of the left had a more determinist view. The regeneration of post-industrial cities gave artists opportunities to secure temporary spaces to make and present work, thereby increasing the cultural capital of the area and its gentrification. In most cases, artists became priced out and were moved on to poorer areas, and the cycle continued. The Arts Council restructuring that formed the RABs was broadly successful despite it being a cost-cutting exercise, as artists and arts organisations had direct contact with knowledgeable ACE officers. Experimental and challenging work that was publicly accessible was encouraged. This enabled artists to gain funding to set up their own spaces and events and for organisations to develop. Culture's importance was recognised politically, and the Government *Department of Culture, Media*, and *Sport* (formed by Labour in 1997) has its own Minister of State with a cabinet position. The tension between freedom of expression and political intervention within the department was generally driven by the ideological imperatives of the party in government.

Modernist thought continued to be the dominant condition within fine art institutions and funding bodies. In art schools, departments whose staff and students set-up *third areas* instigated change, with conceptual and time-based art eventually becoming integrated into academic programmes. A majority of art schools lost their autonomy, becoming polytechnics around 1970. And after 1992, the divide between universities and polytechnics ended. Art courses became subsumed into the faculties of the new semi-private (ex-poly) universities. The shift from block grants to student fees (where income followed the student) contributed to the marketisation of higher education. Senior managers who recognised the requirements for creative practice and related pedagogies enabled courses to flourish, while others questioned the need for expensive technical resources, studio space and high levels of staff in relation to student numbers. The system of fees and loans was alien to prospective working-class, black, and minority ethnic students and the consequence was that the expansion of higher education came largely from the middle classes. Fine art, art history and aesthetics had generally been separate, with fine art being in the main practical and within art schools and art history within art departments of (pre-1992) universities (and in a few institutions coupled with fine art). Aesthetics was generally outside of fine art departments but generally found in philosophy faculties. The conflict within art history—a biographical canon of isms—became threatened by feminist, sociological, post-structural and post-colonial critique. Contemporary aesthetics (art theory) effectively replaced traditional art history in progressive fine art courses. Conversely, art criticism within journalism became reduced to opinion and commentary, reflecting novelty, financial impact, laudable purpose, or populist outrage.

It is evident from this research that live and interdisciplinary art in the UK was, in the late sixties and early seventies, a rejection of and a challenge to dominant cultural codes

and forms. The development of this work was facilitated through access to art education, and that education included: (i) the availability of facilities—space, time, materials and equipment; (ii) the promotion of contemporary art and its contexts; and (iii) connections to artists and arts organisations external to the art school. It had generally been the case that students were often more knowledgeable than their tutors in respect of contemporary trends and reacted against perceived conservatism. There were limited opportunities for experimental and performance artists until the late 1970s, when artists began to organise festivals and events. Magazines and television programmes proliferated and gave access to different forms of expression and thought. Funding bodies became supportive, application processes were more straightforward and artists were paid. Collaborations between art schools, galleries, and independent organisations enabled regular programmes to take place and there was a more international outlook for events and festivals. Despite government cutbacks in the 1980s, the *arms-length* approach to the Arts Council ensured that there was no interference by the government in what was being funded. Local government and the BBC, however, were affected by Conservative governments by way of subtle threats, severe cuts in funding, and, in the case of the Greater London Council, South Yorkshire Council, and others, abolishment.

## 5. Not Concluded

The emphasis of this article has been on the later years of the twentieth-century, a period when cultural avant-gardes shifted from progressive experimental art-forms or disciplines towards the novel (*cutting-edge*?) and redefined as *contemporary art*. Although the combination of presence and duration gives live and other time-based work its potency and offers some resistance to commodification, there are still tensions between art and entertainment, the intelligible and the sensible, the live and recorded, the integral and the supplementary, originality and reproduction and so on. These factors are rarely addressed through object-based work that does not engage with duration, participation, or audience, even though the process of looking or listening is temporal in itself.

In the 1980s, a right-wing government cut and privatised public services, reduced wages and challenged the necessity of publicly funded arts. It was during this period that more radical arts practices proliferated, reflecting a more general opposition to neo-liberal ideology. Then there were confluences between students and lecturers, artist-run spaces, municipal galleries and arts centres. It was by necessity that grass-roots arts organisations developed and that most of these were concerned with live and interdisciplinary art—art that was public, temporal, and often place-specific. There are correlations now with the circumstances of thirty years ago, but as the gallery became a shop, a tourist attraction, or a space for showing research outcomes, a critically or sensory-engaged artist is compelled to operate outside of these contexts.

This century has seen an increasing number of artists working through a range of socially engaged or participatory art practices. Claire Bishop, in *Artificial Hells*, says that the ground of participatory art

> has shifted over the course of the twentieth century, so the identity of participants has been reimagined at each historical moment: from a crowd (1910s), to the masses (1920s), to the people (late 1960s/1970s), to the excluded (1980s), to community (1990s), to today's volunteers whose participation is continuous with a culture of reality television and social networking [2010s]. (Bishop 2012, p. 277)

Speculatively, to update the above, the identity of those participatory artists since 2010 may be the disillusioned, with activity moving more towards political, social, and environmental practices, campaigns and protests. The connections between participatory and performance art are essentially those of presence and engagement, whether physical or cerebral.

> what I value is a sense of actuality, in the strong sense of the term: artworks that are able to constellate not only different registers of experience (aesthetic, cognitive and critical) but also different orders of temporality.... actuality cannot

be fixed on a traumatic view of the past: that is, even as it calls up past art, it must also open onto future work. (Foster 2017, pp. 139–40)

Such work may be closer to environmental actions, community projects, social work, or other socially engaged practices. Ideas of art-form, discipline, etc., are not relevant in this respect and, in fact, no longer exist. When real life intervenes financial circumstances become difficult, creative and political actions take place through necessity and in one's own spare-time. It is the confluence of motivated young artists (without material means) with those committed individuals connected to institutions (that have resources) that can potentially develop the art necessary for these times. The need is for a shared experience of art through unmediated presence, liveness, temporalit and duration.

**Further Information**

Artsadmin. https://www.artsadmin.co.uk/

*Fast and Loose (my dead gallery). 1956–2006.* https://www.fieldgategallery.com/fast-and-loose

History of Lux. https://lux.org.uk/about/our-history/

Live Art Archive. http://www.bristol.ac.uk/theatre-collection/explore/live-art/

Live Art Development Agency. https://www.thisisliveart.co.uk/

*Live Art in the UK: Lois Keidan interview part1* del Rey, Eva 1 British Library. https://blogs.bl.uk/sound-and-vision/2017/04/live-art-in-the-uk-lois-keidan-interview-part-1-of-2.html

Performance Space. https://performancespace.org/

Transmission Gallery, Glasgow. https://transmissiongallery.org/

**Funding:** This research received no funding.

**Data Availability Statement:** No new data were created in this study. Data sharing is not applicable to this article.

**Conflicts of Interest:** The author declares no conflict of interest.

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
