# Peer review of "Performance, Art, Institutions and Interdisciplinarity"

_arts, 1979_

Round 1
Reviewer 1 Report
Comments and Suggestions for Authors
The primary strength of this article rests in its density of often first-hand detail and its unpretentious and materialist method. Its primary weakness is in a somewhat aleatory structure where the flow of ideas is hard to follow and the relationship of detail to idea (or evidence to claim) is difficult to discern.
I take the primary argument to be that the underground arts actually fluorish during circumstances of privation because the official organs of support haven't yet warped a community into sycophants or teachers or grant-writers.
For this claim (if I am indeed right that this is the claim) to be made more persuasive, the historical timeline here needs to be made clearer. In the current draft we jump back and forth in time at a dizzying rate. I recommend starting with the 1979 elections and then the 1986 construction of the Combined Arts panel to help ground and periodize the argument.
In terms of method, the author mentions personal experience informing the research and this gives the writing a welcome on-the-ground flavor in touch with the confused, everyday experience of the many artists themselves. I am especially sympathetic to both the personal POV and the materialist (base-superstructure) impulse here, but the personal tone often means statements are impressionistically sweeping and lacking either more precise qualifiers, citations or examples to make them persuasive. (e.g. The mainstream media was still ridiculing Picasso, Pop art and the abstract expressionists even into the 1970s.) The flip-side of sweeping statements that could be better substantiated are the paragraphs that begin with historical events rather than the main idea the events serve to illustrate.
I want to be clear that I think this article--if restructured and rewritten--promises to be an exciting contribution to the scholarship of the avant garde multi/inter-media performance art scene in the UK since the advent of punk and 80s underground movements. I do think it needs a substantial rewrite to make the most of the author's own wealth of research and experience.
where the funding conditions for the arts are seen as central; but I often had difficulties following the flow of ideas from paragraph to paragraph or section to section. The historical scope should be made more precise and better organized; in the present draft it jumps around at a pace that is hard to follow.
Author Response
Thank you for your review which I appreciate and have considered carefully. To address your main issue I have amended the manuscript by writing a new introduction that outlines the structure and the purpose of each section. Additionally I have included a new section that summarises the findings of the research. These additions make the connection of detail to idea more explicit. I have maintained the largely thematic and nonlinear structure, reflecting the circumstances of the period in question, to avoid a deterministic narrative. It is attempting to keep a balance between a more open text with factual clarity while being more discursive. I recognise that there was some lack of clarity in my arguments regarding your point about the flourishing of underground arts. I did not want to claim that privation caused this but to outline the various circumstances that produced change. I have amended the specific sentence that had given that impression.
I have amended the text to substantiate points and deleted what may have been thought of as sweeping statements. I have ensured that all my sources and citations have now been adequately and consistently referenced. I have rewritten a few more sections and adjusted the structure in places.
Reviewer 2 Report
Comments and Suggestions for Authors
The article is relevant, however, in my opinion, there could be more data to prove the statements and a starting point and a clearer conclusion in scientific terms.
Author Response
Thank you for your review.
I have made some amendment including a new introduction that outlines intention and structure and a new section that summarises the findings of the research.
There are small changes, corrections and additions to the text which gives it clarity.
I have ensured that all my sources and citations have now been adequately and consistently referenced.
Reviewer 3 Report
Comments and Suggestions for Authors
Check application of the standard APA.
Insert page number in the quotes when missing - eg. line 31.
Review graphic aspects: eliminate double spaces between words and/or beginning of sentences - see the entire paper.
Standardize writing: lowercase and uppercase letters - review the entire paper.
View inserted note regarding authors' incomplete reference > eg. Alfred Jarry instead of "Jarry"...line 411.
Complete the reference of artist's works > eg. "Gormley's Angel of the North" (insert date) > line 221.

Author Response
I have checked and amended my referencing to be consistent, correct and with page numbers and accurate identification for online sources. I have standardised lower & upper case letters.
The spacing is caused by mdpi’s design - the text is justified but comparatively narrow.I have made some amendment including a new introduction that outlines intention and structure and a new section that summarises the findings of the research.
Round 2
Reviewer 1 Report
Comments and Suggestions for Authors
The additions to the previous draft are significant and reflect honest, good-faith engagement with peer reviewer comments. I believe the new draft represents a significant and exciting expansion, especially the additions in the body and the conclusion. In particular I appreciated the heuristic of the modernism/post-modernism split to help elucidate the various interpretive schema applied by media and academic coverage of the UK arts community. My only suggestion for subsequent revision before publication would be to rewrite the "Introduction" paragraph at the start of the essay. Though this provides a roadmap of the article it uses a somewhat convoluted passive-voice construction that I found off-putting and out of step with the best parts of the article when the language is clearer and conceptual stakes more straightforward. I would suggest borrowing from or moving the conceptual schema from the added "Summary" section to the Introduction to give context, grounding, and readerly buy-in (motivation) before launching into a blow-by-blow roadmap.
Author Response
Thank you for your response, really appreciated. I concur completely with your comment and recommendation for the introduction. I have added a couple of paragraphs to give more context and have amended and extended the ‘road map’ so it flows better. I’ve corrected some errors and made some small additions. I think it is much improved.